# Diagnostic and prognostic roles of circulating miRNA-223-3p in hepatitis B virus–related hepatocellular carcinoma

Pornpitra Pratedrat[1], Natthaya Chuaypen[1], Pattaraporn Nimsamer[2], Sunchai Payungporn[2], Nutcha Pinjaroen[3], Boonchoo Sirichindakul[4], Pisit Tangkijvanich[1]*

1 Center of Excellence in Hepatitis and Liver Cancer, Faculty of Medicine, Chulalongkorn University, Bangkok, Thailand, 2 Department of Biochemistry, Faculty of Medicine, Chulalongkorn University, Bangkok, Thailand, 3 Department of Radiology, Faculty of Medicine, Chulalongkorn University, Bangkok, Thailand, 4 Department of Surgery, Faculty of Medicine, Chulalongkorn University, Bangkok, Thailand

* pisittkvn@yahoo.com

**Data Availability Statement:** All relevant data are within the manuscript and its Supporting Information files.

## Abstract

### Background

Circulating microRNAs (miRNAs) have been shown to dysregulate in many cancer types including hepatocellular carcinoma (HCC). The purpose of this study was to examine the potential diagnostic or prognostic roles of circulating miRNAs in patients with hepatitis B virus (HBV)-related HCC.

### Methods

Paired cancerous and adjacent non-cancerous liver tissue specimens of patients with HBV-related HCC were used as a discovery set for screening 800 miRNAs by a Nanostring quantitative assay. Differentially expressed miRNAs were then examined by SYBR green quantitative RT-PCR in a validation cohort of serum samples obtained from 70 patients with HBV-related HCC, 70 HBV patients without HCC and 50 healthy controls.

### Results

The discovery set identified miR-223-3p, miR-199a-5p and miR-451a significantly lower expressed in cancerous tissues compared with non-cancerous tissues. In the validated cohort, circulating miR-223-3p levels were significantly lower in the HCC group compared with the other groups. The combined use of serum alpha-fetoprotein and miR-223-3p displayed high sensitivity for detecting early HCC (85%) and intermediate/advanced stage HCC (100%). Additionally, serum miR-223-3p had a negative correlation with tumor size and BCLC stage. On multivariate analysis, serum miR-223-3p was identified as an independent prognostic factor of overall survival in patients with HCC. In contrast, circulating miRNA-199a-5p and miR-451a did not show any clinical benefit for the diagnosis and prognostic prediction of HCC.

**Funding:** Research reported in this publication was supported by the Thailand Research Fund (RTA6280004), the Grant for Chula Research Scholar (CU-GRS-60-06-30-03), the Royal Golden Jubilee Ph.D. Program (Grant No. PHD/0061/2559) and Center of Excellence in Hepatitis and Liver Cancer, Faculty of Medicine, Chulalongkorn University. The funders had no role in study design, data collection and analysis, decision to publish, or preparation of the manuscript.

**Competing interests:** The authors declare that there is no conflicts of interest.

## Conclusions

Our results demonstrated that miR-223-3p was differentially expressed in cancerous compared with paired adjacent non-cancerous tissues. In addition, circulating miRNA-223-3p could represent a novel diagnostic and prognostic marker for patients with HBV-related HCC.

## Introduction

Hepatocellular carcinoma (HCC) represents the sixth most common cancer and the second leading cause of cancer-related mortality worldwide [1]. The prevalence of HCC differs among geographical distribution with its high rate in East and Southeast Asia [2]. In Thailand, major etiologic factors of HCC development are hepatitis B virus (HBV), followed by hepatitis C virus (HCV) infection, alcoholic liver disease (ALD) and non-alcoholic fatty liver disease (NAFLD). Our previous report has demonstrated that at least 60% of HCC cases are attributed to chronic HBV infection [3]. Despite advanced improvements in curative treatment including surgical and ablative therapies, the overall survival of patients with HCC remains unsatisfactory due to tumor aggressiveness at presentation and high rates of recurrence [2]. Currently, prognostic markers for HCC in routine clinical practice are not yet available. Thus, a reliable serum biomarker for early diagnosis, monitoring tumor progression and prognostic prediction is highly needed and would be of great clinical benefit.

Recent advances have demonstrated that the development of HCC is a complex process involving several genetic and epigenetic changes accumulated through repeated hepatocyte destruction and regeneration. Among these alterations, aberrant expression of microRNAs (miR-NAs) plays an essential role in multistep processes of HCC development [4]. MiRNAs are small non-coding RNAs of approximately 18–24 nucleotides that modulate gene expression process by binding to specific mRNA targets and promoting their degradation and translational inhibition [5]. In addition, miRNAs participate in several biological and pathological processes, including cell differentiation, growth, angiogenesis and immune activation [6]. Moreover, it has been shown that miRNA alterations can act as either oncogene activation or tumor suppressor gene inactivation in HCC development [7]. Since dysregulation of miRNA expression has been linked to hepatic carcinogenesis and progression, detecting these miRNAs alterations in liver tissue and blood samples could be potential biomarkers for diagnosis and prognosis of HCC [4, 7]. Although previous studies revealed clinical correlations of miRNA detection in HCC, some reports were limited by insufficient number of screened miRNAs, inadequate independent validation and failure to differentiate HCC from non-HCC chronic liver disease. In addition, liver tissue derived miRNA from liver biopsy might not be ideal biomarkers for HCC because of its invasive procedure with potential complications [8]. Thus, detecting circulating miRNAs in body fluids such as serum specimens is more beneficial for the diagnosis and prognosis of HCC [9].

In this study, we investigated global miRNA expression profiling in liver tissue specimens obtained from patients with HBV-related HCC by NanoString quantitative platform. Additionally, we validated clinical significance of candidate miRNAs by Real-Time PCR in another set of serum specimens obtained from patients with HBV-related HCC.

## Materials and methods

### Patients and sample collection

To investigate miRNA expression profiles in liver tissue, pairs of cancerous and adjacent non-cancerous tissue specimens were obtained from patients with HBV-related HCC undergoing

surgical resection at the Department of Surgery, Chulalongkorn University from 2013 to 2017. The diagnosis of HCC was confirmed by histopathology. Liver tissue samples were immediately stored in RNA later solution (Cat No. 76106, QIAGEN, and Germany) and were then frozen in liquid nitrogen until RNA extraction.

For the validation cohort, stored serum samples were collected from patients with HBV-related HCC, who were diagnosed for the first time at King Chulalongkorn Memorial Hospital, Bangkok, Thailand between 2013 and 2017. The diagnosis of HCC in this cohort was based on typical imaging studies and/or histopathology in accordance with the American Association for the Study of Liver Diseases (AASLD) guideline [10]. Briefly, the criteria of HCC diagnosis were included focal hepatic lesions with hyperattenuation at the arterial phase and hypoattenuation at the portal phase in dynamic computed tomography (CT) or magnetic resonance imaging (MRI). Liver biopsy or fine needle aspiration were performed to confirm HCC diagnosis in cases without typical imaging features. Baseline clinical parameters of these patients were collected, which included gender, age, biochemical liver function tests, Child-Pugh classification, serum alpha-fetoprotein (AFP) and staging of HCC classified by the Barcelona Clinic Liver Cancer (BCLC) [11]. In addition, overall survival (OS) of the patients characterized by the interval between initial recruitment and death or the last follow-up visit was recorded.

Stored serum samples were also obtained from two additional groups, including healthy blood donors with no apparent liver disease (healthy controls) and patients with chronic HBV infection who did not have evidence of HCC (the non-HCC group). Healthy controls and the non-HCC group were matched for gender and age with patients with HCC to reduce confounding effects. The diagnosis of chronic HBV infection was confirmed by the presence of serum hepatitis B s antigen (HBsAg) for at least 6 months. Patients with HCV and/or human immunodeficiency virus (HIV) co-infection were excluded. Collected serum sample specimens were separated by centrifugation and stored at -70˚C until further tested.

This study was performed according to the Declaration of Helsinki for the participation of human individuals. The protocol of this study had been approved by the Institute Ethics Committee of Faculty of Medicine, Chulalongkorn University (IRB No. 726/60) and written informed consent was obtained from each participant.

## Nanostring nCounter analysis

Total RNAs extracted from 25 mg of HCC frozen tissues and matched adjacent non-cancerous tissues individually were extracted for miRNAs using the GenUP™ Total RNA Kit (Biotechrabbit GmbH, Hennigsdorf, Germany). These RNAs were determined for total 800 miRNA expression using the Human nCounter miRNA Assay v 3.0 kit following the manufacturer's instructions (NanoString Technologies, Seattle, WA, USA). Candidate miRNAs were selected if their differential expression in cancerous and non-cancerous tissues were more than 5-folds.

## Validation of NanoString results by qRT-PCR analysis

From NanoString analysis, we selected candidate miRNAs that were significantly differential expression between cancerous and non-cancerous tissues. After filtering, the top 3 candidate miRNAs, including miR-223-3p, miR-199a-5p and miR-451a, were selected for validate their circulating expression levels by qRT-PCR analysis. Briefly, 200 μL of serum samples were extracted using a miRNA isolation kit (Cat No. RMI050, Geneaid Biotech, USA). For cDNA synthesis and poly (A) tail addition, the first-strand cDNA was then synthesized from purified total RNA using SL-poly (A) sequence: GTCGTATCCAGTGCAGGGTCCGAGGTAT TCGCA CTGGATACGACAAAAAAAAAAAAAAAAAAVN and RevertAid First Strand cDNA Synthesis

Kit (Cat No. 1622, Thermo Scientific, USA). The candidate miRNAs were assessed in duplicate by quantitative RT-PCR using SYBR Green system (QPCR Green Master Mix HRox, Cat No. BR0500402, Biotechrabbit, Germany) with specific primers including miR-223-3p:5′ TGTC AGTTTGTCAAATACCCCA3′ (forward),miR-199a5p:5′ AGTGTTCAGAC TACCTGTTC3′ (forward), miR-451a: 5′ AAACCGTTACCATTACTGAGTT3′ (forward) and SL-polyA-R: 5′ GCAGGGTCCGAGGTATTCG3′ (universal reverse). MiR-21 was used as the internal control [12]. Downregulation was confirmed by StepOnePlus Real-time PCR System (Applied Biosystems, USA). The relative expression level was calculated by ΔΔCT of the $2^{-\Delta\Delta CT}$ method [13].

## Statistical analysis

NanoString analysis was performed using nSolver Analysis Software (Version 4.0). Data according to miRNAs differential expressions levels were analyzed with SPSS statistics version 22 (SPSS Inc., Chicago, IL) and GraphPad Prism 5.0 (GraphPad software, CA, USA) software. Quantitative data were calculated in terms of mean ± standard and percentages. Comparisons between groups were analyzed by $\chi^2$ or Fisher's exact test for categorical variables and by Student's *t*-test for quantitative variables. Correlations between baseline parameters were analyzed by the Spearman's rank test. Survival curves were calculated based on Kaplan-Meier analysis and log-rank test. The Cox regression analysis was conducted to identify independent baseline factors associated with overall survival (OS) of patients with HCC. *P* values <0.05 were indicated statistical significance.

## Results

### Expression profiling of microRNAs in liver tissues

A discovery set comprised of 4 pair cancerous and surrounding non-cancerous tissue specimens obtained from 4 patients with HBV-related HCC undergoing surgical resection. All these patients had early stages (BCLC stage 0 or A) and did not receive any therapy for HCC before collecting the samples (S1 Table). Based on the NanoString platform, expression levels of 800 microRNAs were obtained. As shown in Fig 1, a volcano plot demonstrated that 21 miRNAs were significantly differentially expressed in cancerous and non-cancerous tissues. After filtering, the top 3 candidate miRNAs, including miR-223-3p, miR-199a-5p and miR-451a, were selected for further validation in serum specimens by qRT-PCR analysis.

### Validation of miRNA ratios by RT-qPCR

To further investigate whether circulating miRNA expression could be applied for diagnosis of HCC; RT-qPCR was performed in serum samples of a validation cohort consisting of 70 patients with HBV-related HCC, 70 patients without HCC and 50 healthy controls. Among these individuals, healthy controls and the non-HCC group were matched for gender and age with patients with HCC (Table 1). Compared with patients without HCC, the HCC group had significantly lower serum albumin and platelet count but higher serum total bilirubin, aspartate aminotransferase (AST), alanine aminotransferase (ALT), AFP level, the presence of HBV genotype C and cirrhosis. There was no significant difference in terms of HBeAg positivity and HBV DNA level. Among the HCC group, the initial cancer stages based on the BCLC criteria were as follows: 21 (30.0%), 24(34.3%) and 25(35.7%) patients in stages 0-A, B and C-D, respectively.

For further analysis, three miRNAs, including miR-223-3p, miR-199a-5p and miR-451a, identified in the discovery setting and known to be dysregulated in HCC were selected. Relative expression miRNAs levels of the subjects in each group are plotted in Fig 2A, 2B and 2C.

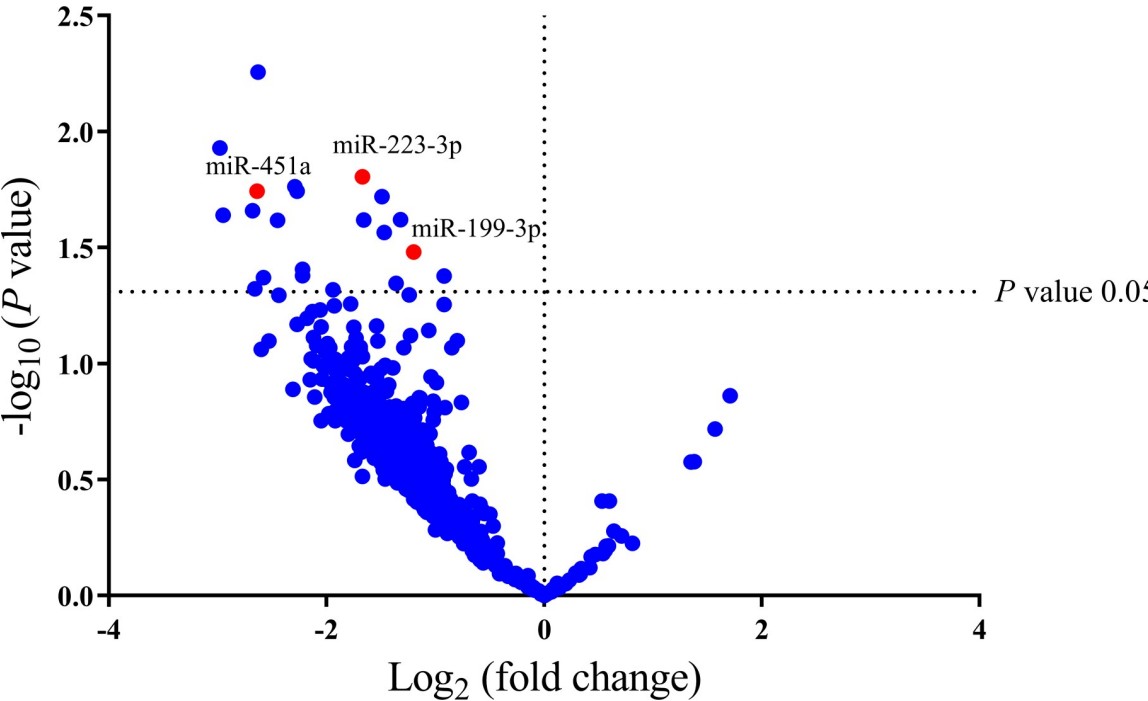

**Fig 1. Volcano plot shows the profile of the differential miRNAs expression in 4 tissue samples of patients with HCC.** The x-axis represents the $Log_2$ fold change and y-axis represents significance level expressed as the—$log_{10}$ $P$-value. The red 3 dots shown in the volcano plot are the downregulated miRNAs in HCC tissue. The blue dots represent miRNAs that did not demonstrate a significant change in expression. Significance value was indicated with a $P$-value cut-off of 0.05.

The results showed that circulating miR-223-3p levels in the HCC group (0.31±0.46) were significantly lower than the non-HCC group (0.94±1.05, $P<0.001$) and healthy controls (1.00 ±0.00, $P<0.001$). In contrast, circulating miR-199a-5p (0.83±1.19) levels in the HCC group

**Table 1. Demographic data of the studied cohorts.**

| Baseline Characteristics | Healthy controls (n = 50) | Patients without HCC (n = 70) | Patients with HCC (n = 70) | P |
|---|---|---|---|---|
| Age (years) | 51.7±4.0 | 51.2±10.9 | 52.0±8.3 | 0.867 |
| Gender (Male) | 30 (60.0) | 42 (60.0) | 51(72.9) | 0.202 |
| Aspartate aminotransferase (IU/L) | | 33.9±30.2 | 116.1±135.9 | <0.001* |
| Alanine aminotransferase (IU/L) | | 37.0±34.0 | 66.4±52.9 | <0.001* |
| Serum albumin (g/dL) | | 4.2±0.4 | 3.6±0.5 | <0.001* |
| Total bilirubin (mg/dL) | | 0.7±0.3 | 1.1±0.7 | <0.001* |
| Platelet count ($10^9$/L) | | 214.0±56.3 | 154.6±70.1 | <0.001* |
| HBeAg positivity | | 29 (41.4) | 24 (34.3) | 0.486 |
| HBV genotype | | | | 0.001 |
| B | | 17 (24.3) | 4 (5.7) | |
| C | | 52 (74.3) | 58 (82.9) | |
| Unknown | | 1 (1.4) | 8 (11.4) | |
| $Log_{10}$ HBV DNA (IU/mL) | | 4.7±2.0 | 4.5±1.7 | 0.491 |
| Alpha fetoprotein (ng/mL) | | 6.4±5.7 | 7877.4±21065.5 | 0.011* |
| Presence of cirrhosis | | 11 (15.7) | 60 (85.7) | <0.001* |
| BCLC stage (0-A/B/C-D) | | - | 21(30.0)/24(34.3)/25(35.7) | - |

Data express as mean±SD and n (%)

(A)
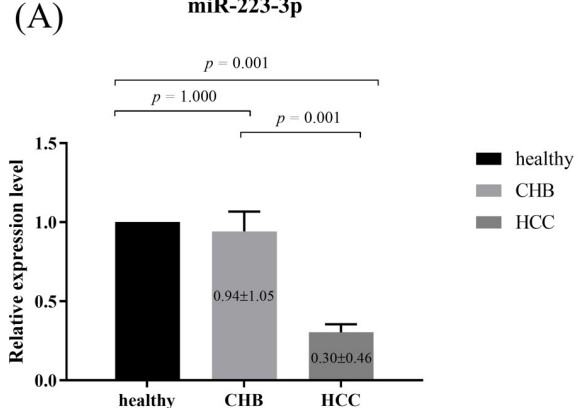

(B)
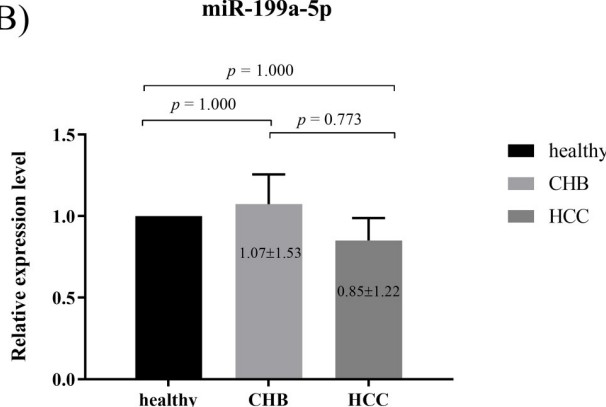

(C)
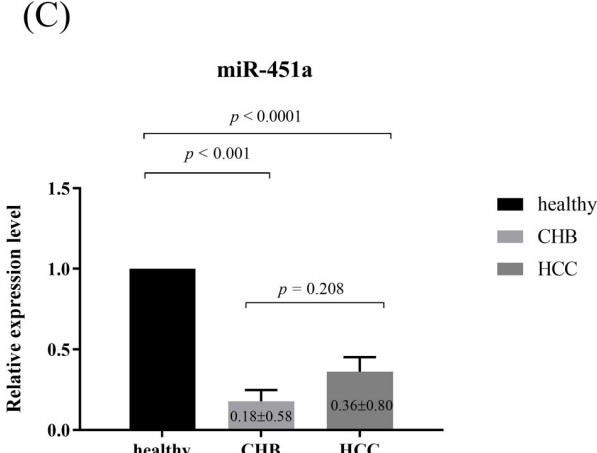

**Fig 2.** Relative expression of serum (A) miR-223-3p, (B) miR-199a-5p and (C) miR-451a in studied groups.

were not significantly different from those of the non-HCC group (1.07±1.53, $P$ = 0.300) and healthy controls (1.00±0.00, $P$ = 0.240). For miR-451a, their expression levels in the HCC group were comparable with the non-HCC group (0.38±0.83 vs 0.18±0.58, $P$ = 0.092) but significantly differed from those of healthy controls ($P$<0.001).

## Relationship between serum miRNAs levels and clinical parameters

Among patients with HCC, the correlation between serum miRNAs levels and clinical parameters at baseline was examined. There was no correlation between miR-223-3p and miR-199a-5p (r = 0.118, $P$ = 0.329), miR-223-3p and miR-451a (r = -0.059, $P$ = 0.28), as well as miR-199a-5p and miR-451a (r = 0.149, $P$ = 0.218). Serum miR-223-3p had a negative correlation with tumor size (r = -0.312, $P$ = 0.008) and BCLC stage (r = -0.410, $P$<0.001) but did not correlated with other clinical parameters, including sex, age, biochemical parameters, AFP level, HBeAg status and HBV DNA level. Serum miR-199a-5p level did not have significant correlation with clinical parameters. For serum miR-451a, its level was negatively correlated with AST (r = -0.308, $P$ = 0.009), ALT (r = -0.381, $P$ = 0.001) and tumor size (r = -0.277, $P$ = 0.020), but did not correlate with other parameters.

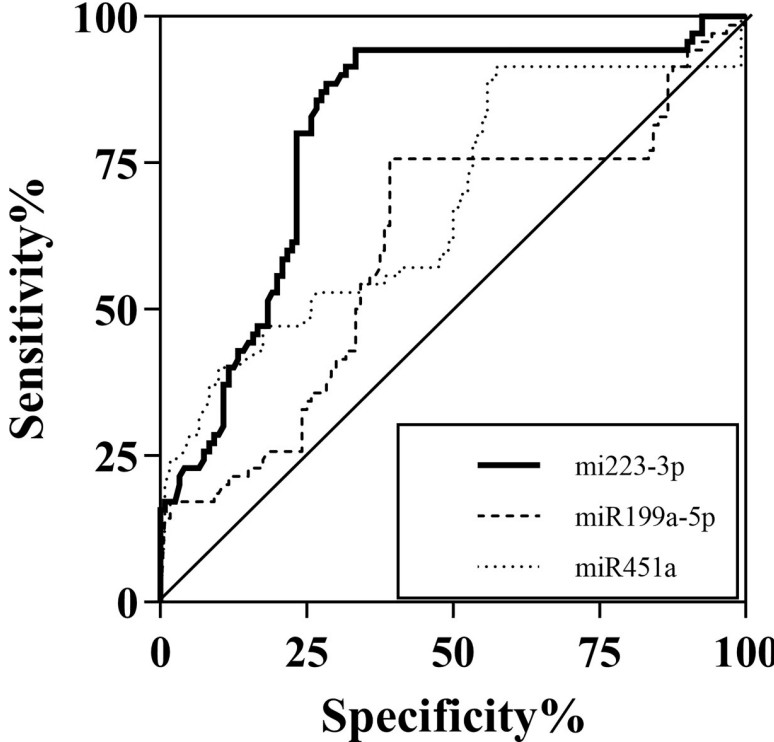

**Fig 3. The area under the curve (AUROC) of HCC and the other groups.**

### Serum miRNAs as diagnostic markers of HCC

The ROC curves for miR-223-3p, miR-199a-5p and miR-451a were generated on the same graph to compare their diagnostic accuracies. As shown in Fig 3, the area under the curve (AUROC) of HCC and the other groups was 0.81 [95% confidence interval (CI); 0.74–0.87, $P<0.001$] for miR-223-3p, 0.61(95%CI; 0.52–0.69, $P = 0.015$) for miR-199a-5p and 0.68 (95% CI; 0.60–0.76, $P<0.001$) for miR-451a. As miR-223-3p was considered the best biomarkers among the studied miRNAs, its diagnostic role was further evaluated. Based on the ROC curve analysis, a cut-off point of serum miR-223-3p levels that exhibited the highest accuracy for differentiating HCC from the other groups was approximately 0.4. At this cut-off level, its sensitivity and specificity were approximately 76.7%, and 80%, respectively.

Regarding all HCC, 56 of 70 (80.0%) patients had decreased levels of serum miR-223-3p $<0.4$, whereas 43 (61.4%) patients had elevated serum AFP at cut-off value of 20 ng/mL. When both serum markers were combined, the sensitivity was increased to 95.7%. For early-stage HCC (stages 0 and A), 13 of 21 (61.9%) patients had decreased levels of serum miR-223-3p $<0.4$, whereas 12 (57.1%) patients had elevated serum AFP at cut-off value of 20 ng/mL. When both markers were used together, the sensitivity of the combined test was 85.7% and 100% for detecting early-stage and intermediate/advanced stage HCC, respectively. The sensitivity of both markers and their combination in relation with BCLC staging is showed in Table 2.

### Serum miRNAs in predicting overall survival of patients with HCC

Apart from clinical correlations, we further examined the potential prognostic role of serum miR-223-3p, miR-199a-5p and miR-451a. Using the median value (0.13) as the cut-off level, the median OS of patients with serum miR-223-3p levels of $\geq$0.13 was 44.4 months, which was

**Table 2. The sensitivity of serum miR-223-3p, AFP and combination in diagnosis of HCC according to tumor staging.**

| BCLC Staging | miR-223-3p (0.4) | AFP (20 ng/ml) | miR-223-3p and AFP |
|---|---|---|---|
| Stage 0, A (n = 21) | 13 (61.9) | 12 (57.1) | 18 (85.7) |
| Stage B (n = 24) | 18 (75.0) | 15 (62.5) | 24 (100) |
| Stage C,D (n = 25) | 25 (100) | 16 (64.0) | 25 (100) |

Data expressed as n (%)

significantly better than that of patients whose serum levels were <1.3 (median OS, 20.7 months; $P$ = 0.001 by log rank test) (Fig 4A). For serum miR-199a-5p, there was no difference in OS between patients with high or low levels (Fig 4B). Likewise, there was no difference in OS between patients with high or low levels of serum miR-451a (Fig 4C).

Serum miR-223-3p, miR-199a-5p and miR-451a levels were entered into the univariate analysis together with other variables that might influence the prognosis of HCC. These baseline factors included age, gender, AST and ALT levels, platelet counts, HBeAg status, HBV DNA level, AFP level, Child classification, tumor size and the BCLC stage. Among these parameters, AFP, tumor size, the BCLC stage and serum miR-223-3p were selected for multivariate analysis. Based on multivariate analysis, the data revealed that independent prognostic factors of OS were the BCLC stage and serum miR-223-3p (Table 3).

## Discussion

Growing evidence has indicated a critical role of miRNAs in hepatocarcinogenesis [4]. It is well recognized that these small non-coding RNAs involve in post-transcriptional regulation of gene expression and many cellular processes such as growth, angiogenesis, differentiation and metastasis[6]. Recent data have demonstrated that miRNA deregulation are already detectable in premalignant lesions including low-grade or high-grade dysplastic nodules, indicative of their contribution in early stages of HCC development [14]. Moreover, deregulated miRNAs are shown to be pivotal modulators in tumor progression and aggressiveness, in which they act as oncogenes or tumor suppressors [7]. Considering their contribution in various stages of hepatocarcinogenesis and their stability in body fluids, it appears that circulating miRNAs could serve as potential diagnostic and prognostic biomarkers for HCC.

The present study was primarily designed to compare miRNAs expression in tissue samples of patients with HBV-related HCC, and then validated the clinical implication of selected candidate miRNAs in serum specimens. In the discovery cohort, we applied the NanoString platform that allowed the detection of tissue miRNAs with high sensitivity and specificity. Based on this technique, we identified 29 miRNAs that were significantly differentially expressed in cancerous and non-cancerous tissues. Among these, miR-223-3p was significantly down-regulated expressed in cancerous tissue compared to their respective controls. Such finding was in agreement with previous data demonstrated that highly deregulated miR-223 expression could markedly distinguish HCC from adjacent non-cancerous liver, irrespective of viral hepatitis status [15, 16]. For instance, repressed miR-223 expression in cancerous tissues obtained from patients with chronic viral hepatitis exhibited median 5-fold reduction compared with the non-cancerous counterpart, while the corresponding figure observed in tumors derived from non-B non-C HCC was approximately 3.5-fold reduction [15]. Thus, altered expression of miR-223-3p in liver tissue suggested that this miRNA could hold value as a novel biomarker for HCC.

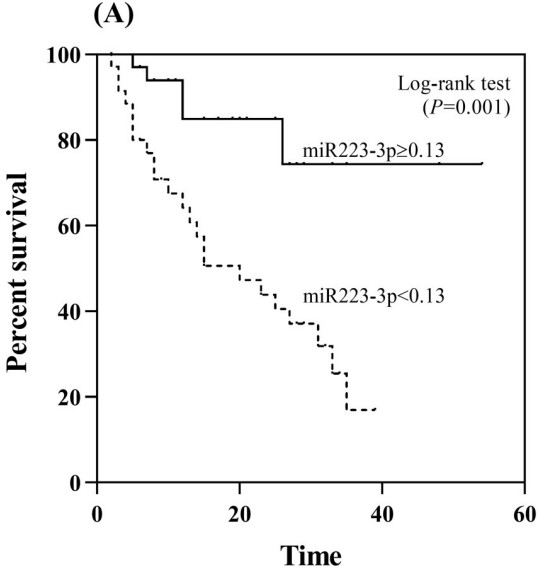

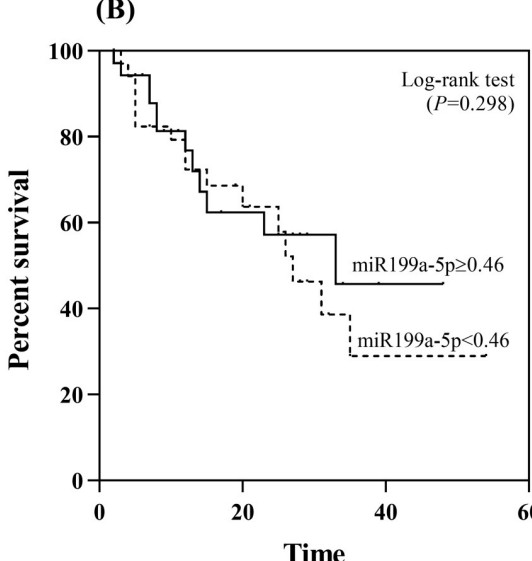

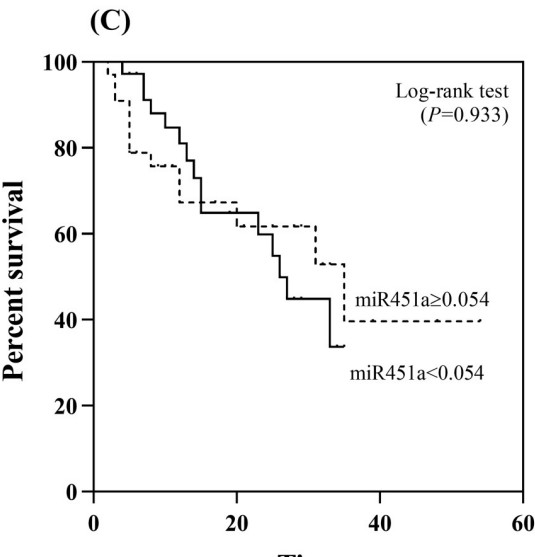

**Fig 4. Association of the studied miRNAs with overall survival in patients with HCC.** (A) miR-223-3p (B) miR-199a-5p (C) miR-451a.

In the validated cohort of circulating miRNAs, we found that serum miR-223-3p levels were significantly lower in patients with HCC compared with the non-HCC group and healthy controls. Based on ROC curve analysis, it was revealed that circulating miR-223-3p could help discriminate HCC from the other two groups. At the cut-off level of 0.4, its sensitivity and specificity were approximately 80%. When this biomarker were applied together with serum AFP, the sensitivity of the combined test was approximately 85% for early cancer and reached 100% for detecting intermediate and advanced stage HCC. Currently, serum AFP is the most widely used tumor marker for the detection and monitoring of HCC. However, this fetal-specific glycoprotein lacks its sensitivity in detecting early HCC [17]. Thus, the combination of AFP and serum miR-223-3p could be considered as a potentially promising tool to better

**Table 3. Variables associated with overall survival in patients with HCC.**

| Variables | Category | Overall survival | | | |
|---|---|---|---|---|---|
| | | Univariate analysis | | Multivariate analysis | |
| | | OR (95%CI) | *P* | OR (95%CI) | *P* |
| Age (years) | < 50 vs. ≥ 50 | 0.70 (0.33–1.48) | 0.347 | | |
| Gender | Male vs. Female | 1.17 (0.51–2.68) | 0.709 | | |
| Aspartate aminotransferase (IU/L) | < 60 vs. ≥ 60 | 1.44 (0.68–3.05) | 0.342 | | |
| Alanine aminotransferase (IU/L) | < 50 vs. ≥ 50 | 0.76 (0.36–1.60) | 0.465 | | |
| Platelet count ($10^9$/L) | ≥ 140 vs. <140 | 1.81 (0.83–3.95) | 0.134 | | |
| HBeAg positivity | Negative vs. Positive | 1.43 (0.65–3.18) | 0.376 | | |
| $Log_{10}$ HBV DNA (IU/mL) | < 4.0 vs. ≥ 4.0 | 0.69 (0.29–1.63) | 0.393 | | |
| Child-Pugh classification | A vs. B and C | 1.26 (0.58–2.76) | 0.560 | | |
| Alpha fetoprotein (ng/mL) | < 100 vs. ≥ 100 | 2.76 (1.26–6.07) | 0.011* | 1.28 (0.55–2.97) | 0.571 |
| Tumor size (cm.) | <5.0 vs. ≥ 5.0 | 4.21 (1.52–11.68) | 0.006* | 0.99 (0.23–4.22) | 0.985 |
| BCLC stage | 0-B vs. C, D | 5.74 (1.86–17.75) | 0.002* | 9.27 (1.99–43.14) | 0.005* |
| miR-223-3p | ≥ 0.13 vs. < 0.13 | 4.38 (1.66–11.56) | 0.003* | 6.61 (2.36–18.55) | <0.001* |
| miR-199a-5p | ≥ 0.46 vs. < 0.46 | 0.81 (0.38–1.72) | 0.589 | | |
| miR-451a | ≥ 0.05 vs. < 0.05 | 1.03 (0.49–2.18) | 0.933 | | |

differentiate HCC from benign liver disorders, as demonstrated in this study. Of note, there was no difference of circulating miR-223-3p levels among non-HCC samples, suggesting that there was no sequential decline of this microRNA from normal controls and patients without HCC. These results were consistent with previous data demonstrating that serum miR-223 levels were significantly decreased in patients with HCC, irrespective of underlying etiologies, compared with sera from individuals without HCC [18–22]. Moreover, a recent meta-analysis indicated that the combination of miRNAs and AFP could significantly increase the diagnostic accuracy compared with the use of each test alone [23]. Interestingly, it was also highlighted in the above-mentioned report that circulating miRNAs exhibited great potential as a novel strategy for early diagnosis of HCC.

Additionally, our results showed that there was a negative correlation between down-regulation of serum miR-223-3p and more aggressive tumor behavior in patients with HCC. Specifically, decreased miR-223-3p level was observed more frequently in patients with large tumor size and advanced BCLC stages. In addition, Kaplan-Meier survival analysis showed that the overall survival of the low serum miR-223-3p group was significantly inferior to that of the high miR-223-3p expression group. Finally, multivariate analysis confirmed that diminished serum miR-223-3p level was an independent, unfavorable predictor of overall survival. To the best of our knowledge, this report is the first to demonstrate the prognostic implication of circulating miR-223-3p in patients with HCC.

As its expression was negatively related to tumor progression, our findings also suggested that this miRNA might display a tumor suppressive role in HCC. In fact, previously *in vitro* studies had reported the tumor suppressive role of miR-223-3p expression in HCC cell lines. For instance, it was shown that downregulation of miR-223 could promote Hep3B cell line via the insulin-like growth factor-1 signaling pathway [24]. In addition, miR-223-223-3p could regulate pyrin domain containing 3 in promoting hep3B apoptosis and inhibit cell proliferation [25]. Similarly, overexpression of miR-223 showed significantly increased HepG2 and Bel-7402 apoptosis and acted as a tumor suppressor through mTOR signaling pathway [26]. Moreover, downregulation of miR-223 could lead to alterations in integrin αV subunit expression, resulting in tumor cell migration and increased metastatic potential [16]. These *in vitro*

studies supported our clinical findings that circulating miR-223 could represent a good prognostic marker for HCC. Regarding miR-199, previous data showed that this miRNA was associated with cell cycle processes in HCC via XBP1/cyclin D and acted as a tumor suppressor by targeting RGS17 in the inhibition of tumor growth [27]. For miR-451, previous published reports suggested that this miRNA strongly repressed HCC proliferation [28–30]. However, our results did not show any clinical significance of miR-199 and miR-451 in terms of diagnosis or prognosis of patients with HCC.

Besides the mentioned miRNAs, other several miRNAs might also involve in various processes of hepatocarcinogenesis [4]. For instance, previous data showed that the let-7 family of microRNAs modulated Bcl-xL expression and potentiated treatment-induced apoptosis in HCC [31]. In the discovery set, we found that hsa-let-7c was significantly up-regulated in cancerous tissues compared with non-cancerous tissues, which was in line with a previous study demonstrating that increased hsa-let-7c expression was associated with HCC progression and poor prognosis [32]. In contrast, other reports identified hsa-let-7c as being downregulated in HCC cell lines and human cancerous tissues [31, 33–34]. As the role of hsa-let-7c in HCC was still uncertain, we decided to exclude the analysis of this miRNA in the validation cohort. Regarding miR-155, previous data revealed a positive correlation between this miRNA and toll-like receptor 7 (TLR7), which might regulate viral transcription during chronic HBV infection [35]. Moreover, in an animal model aberrant expression of miR-155 was found at early stages of NAFLD-induced hepatocarcinogenesis [36]. In our study, however, its expression levels in cancerous and non-cancerous tissues were comparable, indicating this miRNA might not participate in the development of HBV-related HCC.

This report had some limitations as being a retrospective cohort, which recruited relatively small number of patients. Moreover, our study included only patients with chronic HBV infection, which might not be applicable to other etiological causes of chronic liver disease, such as chronic HCV infection, ALD and NAFLD. Despite these limitations, our data clearly demonstrated that miR-223-3p was differentially expressed in cancerous tissues compared with paired adjacent tissues. In addition, circulating miRNA-223-3p levels could differentiate HCC from the non-HCC groups. Apart from its diagnostic role, this circulating miRNA could also serve as a promising prognostic biomarker of HBV-related HCC. Further studies are worthwhile to confirm these observations and to elucidate the clinical significance of serum miRNA-223-3p in patients with HCC.

## Supporting information

**S1 Table. Clinicopathological features of 4 patients undergoing surgical resection.**
(DOCX)

## Acknowledgments

The authors wish to thank all of member in Center of Excellence in Hepatitis and Liver Cancer, Faculty of Medicine, Chulalongkorn University for kind help and good suggestion.

## Author Contributions

**Conceptualization:** Pisit Tangkijvanich.

**Formal analysis:** Natthaya Chuaypen, Sunchai Payungporn.

**Methodology:** Pornpitra Pratedrat, Natthaya Chuaypen, Pattaraporn Nimsamer, Sunchai Payungporn.

**Resources:** Sunchai Payungporn, Nutcha Pinjaroen, Boonchoo Sirichindakul.

**Supervision:** Sunchai Payungporn, Boonchoo Sirichindakul, Pisit Tangkijvanich.

**Validation:** Pornpitra Pratedrat, Pattaraporn Nimsamer, Nutcha Pinjaroen, Boonchoo Sirichindakul.

**Visualization:** Natthaya Chuaypen.

**Writing – original draft:** Pornpitra Pratedrat, Pisit Tangkijvanich.

**Writing – review & editing:** Pisit Tangkijvanich.

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
