## [Decision Letter · Decision Letter 0]

6 Nov 2019

PONE-D-19-28223

Diagnostic and prognostic roles of circulating miRNA-223-3p in hepatitis B virus–related hepatocellular carcinoma

PLOS ONE

Dear Dr. Tangkijvanich,

Thank you for submitting your manuscript to PLOS ONE. After careful consideration, we feel that it has merit but does not fully meet PLOS ONE’s publication criteria as it currently stands, however it generated high enthusiam. Therefore, we invite you to submit a revised version of the manuscript that addresses the points raised during the review process.

We would appreciate receiving your revised manuscript by Dec 21 2019 11:59PM. To enhance the reproducibility of your results, we recommend that if applicable you deposit your laboratory protocols in protocols.io, where a protocol can be assigned its own identifier (DOI) such that it can be cited independently in the future. For instructions see: http://journals.plos.org/plosone/s/submission-guidelines#loc-laboratory-protocols

We look forward to receiving your revised manuscript.

Kind regards,

Ratna B. Ray

Academic Editor

PLOS ONE

Journal Requirements:

Research reported in this publication was supported by the Thailand Research Fund (RTA6280004), the Grant for Chula Research Scholar (CU-GRS-60-06-30-03), the Royal Golden Jubilee Ph.D. Program (Grant No. PHD/0061/2559) and Center of Excellence in Hepatitis and Liver Cancer, Faculty of Medicine, Chulalongkorn University.

The authors have no any financial to disclose. 

Reviewers' comments:

Reviewer's Responses to Questions

**Comments to the Author**

1. Is the manuscript technically sound, and do the data support the conclusions?

Reviewer #1: Partly

Reviewer #2: Yes

2. Has the statistical analysis been performed appropriately and rigorously? 

Reviewer #1: I Don't Know

Reviewer #2: Yes

3. Have the authors made all data underlying the findings in their manuscript fully available?

Reviewer #1: No

Reviewer #2: Yes

4. Is the manuscript presented in an intelligible fashion and written in standard English?

Reviewer #1: Yes

Reviewer #2: Yes

5. Review Comments to the Author

Reviewer #1: Pratedrat P et al. examined the expression of miRs from paired cancerous and adjacent non-cancerous liver tissue specimens of 4 patients with HBV related HCC.

1. Authors should describe which company made “GenUP™ Total RNA Kit”.

2. How did authors measure the miRs from sera? Authors should describe them. Which miR did they use? Because it is difficult to use miR-21 as internal control from sera. Did authors use a spike control? Cf) Nakamura M, et al. Serum microRNA-122 and Wisteria floribunda agglutinin-positive Mac-2 binding protein are useful tools for liquid biopsy of the patients with hepatitis B virus and advanced liver fibrosis. PLoS One. 2017 May 5;12(5):e0177302. doi: 10.1371/journal.pone.0177302.

3. Authors examined the expression of miRs from paired cancerous and adjacent non-cancerous liver tissue specimens of 4 patients with HBV related HCC. Please describe the backgrounds of these four patients in detail. Please demonstrate all data of miRs from these 4 patients.

4. In Table 1, which do they have HBV genotypes?

Reviewer #2: The study has been well designed and it compares miRNAs expression in tissue samples of patients with HBV related HCC and then validated the clinical implication of the selected miRNA s in serum specimens. In the validated cohort of circulating miRNAs it was observed that serum miRNA 223-3P levels were significantly lower in patients with HCC compared with the non HCC group and healthy controls. The combination of AFP and serum miRN-223P could be considered as a potentially promising tool to better differentiate HCC from benign liver disorders. There was also a negative correlation between down regulation of serum MIR-223-3P and aggressive behavior in patients with HCC. The study also demonstrates the prognostic implication of circulatory MIR-223-3P in patients with HCC.

6. PLOS authors have the option to publish the peer review history of their article (what does this mean?). If published, this will include your full peer review and any attached files.

Reviewer #1: No

Reviewer #2: No

---

## [Author Response · Author response to Decision Letter 0]

20 Dec 2019

Dear the editors

We appreciate the opportunity to submit a revision of our manuscript entitled “Diagnostic and prognostic roles of circulating miRNA-223-3p in hepatitis B virus-related hepatocellular carcinoma” for publication in PLOS ONE. It is gratifying that the editors and reviewers consider our study has been well designed and has generated high enthusiasm.

This revised manuscript has been improved as a result of changes made in response to the comments of the reviewers. We have attached a point-by-point response to the comments and have highlighted the changes from the previous version with red color in the revised manuscript. 

This manuscript is not currently under consideration elsewhere, and all authors have approved the submission of this manuscript for publication in PLOS ONE.

Thank you for considering this revised manuscript. We look forward to your kind reply.

Best regards,

Pisit Tangkijvanich, M.D.

Center of Excellence in Hepatitis and Liver Cancer, 

Faculty of Medicine, Chulalongkorn University, 

Bangkok, 10330 Thailand

Telephone: +662-256-4482, Fax: +662-256-4482

E-mail address: pisittkvn@yahoo.com

Comments to the Author

1. Is the manuscript technically sound, and do the data support the conclusions?

Reviewer #1: Partly

Reviewer #2: Yes

2. Has the statistical analysis been performed appropriately and rigorously?

Reviewer #1: I Don't Know

Reviewer #2: Yes 

3. Have the authors made all data underlying the findings in their manuscript fully available?

Reviewer #1: No

Reviewer #2: Yes

4. Is the manuscript presented in an intelligible fashion and written in standard English?

Reviewer #1: Yes

Reviewer #2: Yes

5. Review Comments to the Author

Point-by-Point Response to the Reviewers’ Comments

Reviewer #1: Pratedrat P et al. examined the expression of miRs from paired cancerous and adjacent non-cancerous liver tissue specimens of 4 patients with HBV related HCC.

1. Authors should describe which company made “GenUP™ Total RNA Kit”.

Ans. Thank you for the comment. “GenUP™ Total RNA Kit” was made by Biotechrabbit GmbH, Hennigsdorf, Germany (In Materials and Methods, page 6, line 122).

2. How did authors measure the miRs from sera? Authors should describe them. Which miR did they use? Because it is difficult to use miR-21 as internal control from sera. Did authors use a spike control? Cf) Nakamura M, et al. Serum microRNA-122 and Wisteria floribunda agglutinin-positive Mac-2 binding protein are useful tools for liquid biopsy of the patients with hepatitis B virus and advanced liver fibrosis. PLoS One. 2017 May 5;12(5):e0177302. doi: 10.1371/journal.pone.0177302.

Ans. We appreciate the comment. We agree that selecting a reliable reference gene is necessary for an accurate measurement of circulating miRNA by RT-qPCR. Based on a recent review by Anfossi S, et al (Clinical utility of circulating non-coding RNAs - an update, Nat Rev Clin Oncol 2018;15:541-63), the current gold standard techniques for normalization of miRNA expression data are still lacking. The more widely used methods are spiked-in exogenous controls or stably expressed miRNAs. For a spike-in control, it has been shown that this method could be associated with technical variation from manual pipetting error (de Ronde MWJ, et al, Study Design and qPCR Data Analysis Guidelines for Reliable Circulating miRNA Biomarker Experiments: A Review. Clin Chem. 2018; 64:1308-18) Based on several algorithms, including GeNorm, Normfinder, BestKeeper, and Comparative DCq, the most stably expressed reference genes in serum of patients with HCC appears to be miR-21 (Tang G, et al. Different normalization strategies might cause inconsistent variation in circulating microRNAs in patients with hepatocellular carcinoma. Med Sci Monit 2015; 21: 617-24). Thus, we selected miR-21 as the reference gene in this study and we found that the levels of miR-21 were consistent in all serum samples similar to the previous report.

3. Authors examined the expression of miRs from paired cancerous and adjacent non-cancerous liver tissue specimens of 4 patients with HBV related HCC. Please describe the backgrounds of these four patients in detail. Please demonstrate all data of miRs from these 4 patients.

Ans. The clinical backgrounds of these four patients in detail are shown in Supplement Table 1. All of the data of miRs are shown in Supplement File.

4. In Table 1, which do they have HBV genotypes?

Ans. HBV genotypes in the HCC and non-HCC groups are now added in Table 1. Genotype C was significantly more common in patients with HCC compared with patients without HCC (In Results, page 8, line 181-182).

Reviewer #2: The study has been well designed and it compares miRNAs expression in tissue samples of patients with HBV related HCC and then validated the clinical implication of the selected miRNA s in serum specimens. In the validated cohort of circulating miRNAs it was observed that serum miRNA 223-3P levels were significantly lower in patients with HCC compared with the non HCC group and healthy controls. The combination of AFP and serum miRN-223P could be considered as a potentially promising tool to better differentiate HCC from benign liver disorders. There was also a negative correlation between down regulation of serum MIR-223-3P and aggressive behavior in patients with HCC. The study also demonstrates the prognostic implication of circulatory MIR-223-3P in patients with HCC.

Ans. Thank you very much for the comments.

---

## [Decision Letter · Decision Letter 1]

26 Feb 2020

PONE-D-19-28223R1

Diagnostic and prognostic roles of circulating miRNA-223-3p in hepatitis B virus–related hepatocellular carcinoma

PLOS ONE

Dear Tangkijvanich,

Thank you for submitting your manuscript to PLOS ONE. After careful consideration, we feel that it has merit but does not fully meet PLOS ONE’s publication criteria as it currently stands. Therefore, we invite you to submit a revised version of the manuscript that addresses the points raised during the review process.

In particular, other important microRNAs involved in HBV infection should be discussed

We would appreciate receiving your revised manuscript by 1 month. To enhance the reproducibility of your results, we recommend that if applicable you deposit your laboratory protocols in protocols.io, where a protocol can be assigned its own identifier (DOI) such that it can be cited independently in the future. For instructions see: http://journals.plos.org/plosone/s/submission-guidelines#loc-laboratory-protocols

We look forward to receiving your revised manuscript.

Kind regards,

Isabelle Chemin, PhD

Academic Editor

PLOS ONE

Reviewers' comments:

Reviewer's Responses to Questions

**Comments to the Author**

1. If the authors have adequately addressed your comments raised in a previous round of review and you feel that this manuscript is now acceptable for publication, you may indicate that here to bypass the “Comments to the Author” section, enter your conflict of interest statement in the “Confidential to Editor” section, and submit your "Accept" recommendation.

Reviewer #1: (No Response)

Reviewer #2: All comments have been addressed

2. Is the manuscript technically sound, and do the data support the conclusions?

Reviewer #1: Partly

Reviewer #2: Yes

3. Has the statistical analysis been performed appropriately and rigorously? 

Reviewer #1: I Don't Know

Reviewer #2: Yes

4. Have the authors made all data underlying the findings in their manuscript fully available?

Reviewer #1: No

Reviewer #2: Yes

5. Is the manuscript presented in an intelligible fashion and written in standard English?

Reviewer #1: Yes

Reviewer #2: Yes

6. Review Comments to the Author

Reviewer #1: Authors found that miR-223-3p was differentially expressed in cancerous compared with paired adjacent non-cancerous tissues. However, the process of selection and other important microRNAs in HBV infection should be discussed. Because authors used small number of patients for screening.

1. It is unclear why the authors exclude hsa-let-7c. Discuss about hsa-let-7c, which is upregulated in your law data.

See: Jiang X, et al. Regulation of microRNA by hepatitis B virus infection and their possible association with control of innate immunity. World J Gastroenterol. 2014 Jun 21;20(23):7197-206.

2. Authors should discuss about miR-155. Ref. Sarkar N, et al. Expression of microRNA-155 correlates positively with the expression of Toll-like receptor 7 and modulates hepatitis B virus via C/EBP-β in hepatocytes. J Viral Hepat. 2015 Oct;22(10):817-27.

3. Authors should describe the patient characteristics of patients for Nanostring nCounter Analysis.

Reviewer #2: I find that the clarifications in response to the comments by the referees have been answered well and incorporated in the revised text of the paper.

7. PLOS authors have the option to publish the peer review history of their article (what does this mean?). If published, this will include your full peer review and any attached files.

Reviewer #1: No

Reviewer #2: No

---

## [Author Response · Author response to Decision Letter 1]

25 Mar 2020

Dear the editors

We appreciate the opportunity to submit a revision of our manuscript entitled “Diagnostic and prognostic roles of circulating miRNA-223-3p in hepatitis B virus-related hepatocellular carcinoma” for publication in PLOS ONE. It is gratifying that the editors and reviewers consider our study has been well designed and has generated high enthusiasm.

This revised manuscript has been improved as a result of changes made in response to the comments of the reviewers. We have attached a point-by-point response to the comments and have highlighted the changes from the previous version with red color in the revised manuscript. 

This manuscript is not currently under consideration elsewhere, and all authors have approved the submission of this manuscript for publication in PLOS ONE.

Thank you for considering this revised manuscript. We look forward to your kind reply.

Best regards,

Pisit Tangkijvanich, M.D.

Center of Excellence in Hepatitis and Liver Cancer, 

Faculty of Medicine, Chulalongkorn University, 

Bangkok, 10330 Thailand

Telephone: +662-256-4482, Fax: +662-256-4482

E-mail address: pisittkvn@yahoo.com

Comments to the Author

1. If the authors have adequately addressed your comments raised in a previous round of review and you feel that this manuscript is now acceptable for publication, you may indicate that here to bypass the “Comments to the Author” section, enter your conflict of interest statement in the “Confidential to Editor” section, and submit your "Accept" recommendation.

Reviewer #1: (No Response)

Reviewer #2: All comments have been addressed

2. Is the manuscripts technically sound, and do the data support the conclusions?

Reviewer #1: Partly

Reviewer #2: Yes

3. Has the statistical analysis been performed appropriately and rigorously?

Reviewer #1: I Don't Know

Reviewer #2: Yes

4. Have the authors made all data underlying the findings in their manuscript fully available?

Reviewer #1: No

Reviewer #2: Yes

5. Is the manuscript presented in an intelligible fashion and written in standard English?

Reviewer #1: Yes

Reviewer #2: Yes

6. Review Comments to the Author

Point-by-Point Response to the Reviewers’ Comments

Reviewer #1: Authors found that miR-223-3p was differentially expressed in cancerous compared with paired adjacent non-cancerous tissues. However, the process of selection and other important microRNAs in HBV infection should be discussed because authors used small number of patients for screening.

1. It is unclear why the authors exclude hsa-let-7c. Discuss about hsa-let-7c, which is upregulated in your law data. See: Jiang X, et al. Regulation of microRNA by hepatitis B virus infection and their possible association with control of innate immunity. World J Gastroenterol. 2014 Jun 21;20(23):7197-206.

Ans. Thank you very much for the comment. Based on our data, multiple let-7 isoforms including 7a, 7b, 7c, 7d, 7e, 7f, 7g and 7i were upregulated in cancerous tissues compared with non-cancerous tissues. However, previous reports regarding the role of let-7c in HCC was inconclusive. For example, a study showed that the upregulation of let-7c expression was associated with the progression and poor prognosis of HCC (ref#31). In contrast, other reports demonstrated that let-7c was downregulated in HCC cell lines and human malignant tissue (ref#32 and 33). (In Discussion, second paragraph, page 16).

Moreover, distinguishing let-7 miRNA family members that differ by a single nucleotide is difficult by conventional qRT-PCR. Other more specific methods such as stem-loop RT-qPCR combined with TaqMan-based real-time quantification and locked nucleic acid (LNA) might improve the detection. However, these techniques are highly dependent on the design of the stem-loop and LNA oligomers, which often requires extensive trial-and-error optimization and more expensive than conventional primers (Benes V, Castoldi M. Methods. 2010; 50(4):244-9). For these reasons, we decide to exclude the analysis of let-7c in the validation cohort. 

2. Authors should discuss about miR-155. Ref. Sarkar N, et al. Expression of microRNA-155 correlates positively with the expression of Toll-like receptor 7 and modulates hepatitis B virus via C/EBP-β in hepatocytes. J Viral Hepat. 2015 Oct;22(10):817-27.

Ans. We appreciate the comment. Although this miRNA exhibited a positive correlation with TLR7 and might regulate the pathogenesis of HBV infection (ref#34), our data showed that its expression levels in cancerous tissues were similar to the non-cancerous specimens. Thus, this miRNA might not play an important role in human HCC. (In Discussion, second paragraph, page 16).

3. Authors should describe the patient characteristics of patients for Nanostring nCounter Analysis.

Ans. The characteristics and clinical backgrounds of 4 patients using in Nanostring nCounter analysis are shown in Supplement Table 1. 

Reviewer #2: I find that the clarifications in response to the comments by the referees have been answered well and incorporated in the revised text of the paper.

Ans. Thank you very much for the comments.

---

## [Decision Letter · Decision Letter 2]

10 Apr 2020

Diagnostic and prognostic roles of circulating miRNA-223-3p in hepatitis B virus–related hepatocellular carcinoma

PONE-D-19-28223R2

Dear Dr. Tangkijvanich

We are pleased to inform you that your manuscript has been judged scientifically suitable for publication and will be formally accepted for publication once it complies with all outstanding technical requirements.

With kind regards,

Isabelle Chemin, PhD

Academic Editor

PLOS ONE

Additional Editor Comments (optional):

Reviewers' comments:

Reviewer's Responses to Questions

**Comments to the Author**

1. If the authors have adequately addressed your comments raised in a previous round of review and you feel that this manuscript is now acceptable for publication, you may indicate that here to bypass the “Comments to the Author” section, enter your conflict of interest statement in the “Confidential to Editor” section, and submit your "Accept" recommendation.

Reviewer #1: All comments have been addressed

2. Is the manuscript technically sound, and do the data support the conclusions?

Reviewer #1: Yes

3. Has the statistical analysis been performed appropriately and rigorously? 

Reviewer #1: I Don't Know

4. Have the authors made all data underlying the findings in their manuscript fully available?

Reviewer #1: Yes

5. Is the manuscript presented in an intelligible fashion and written in standard English?

Reviewer #1: Yes

6. Review Comments to the Author

Reviewer #1: (No Response)

7. PLOS authors have the option to publish the peer review history of their article (what does this mean?). If published, this will include your full peer review and any attached files.

Reviewer #1: No

---

## [Editor Report · Acceptance letter]

14 Apr 2020

PONE-D-19-28223R2 

Diagnostic and prognostic roles of circulating miRNA-223-3p in hepatitis B virus–related hepatocellular carcinoma 

Dear Dr. Tangkijvanich:

I am pleased to inform you that your manuscript has been deemed suitable for publication in PLOS ONE. Congratulations! Your manuscript is now with our production department. 

With kind regards,

on behalf of

Mrs Isabelle Chemin 

Academic Editor

PLOS ONE